# KAE: Kolmogorov-Arnold Auto-Encoder for Representation Learning

## Abstract

The Kolmogorov-Arnold Network (KAN) has recently emerged as a promising alternative to traditional multi-layer perceptrons (MLPs), offering enhanced accuracy and interpretability through learnable activation functions on edges instead of fixed functions on nodes. In this paper, we present the Kolmogorov-Arnold Auto-Encoder (KAE), a novel integration of KAN with autoencoders (AEs) that aims to improve representation learning and performance in retrieval, classification, and denoising tasks. By utilizing the flexible polynomial functions in KAN layers, KAE effectively captures complex data patterns and non-linear relationships, outperforming standard autoencoders. Our extensive experiments on benchmark datasets show that KAE significantly enhances the quality of latent representations, resulting in reduced reconstruction and denoising errors, and also improves performance in downstream tasks, including higher classification accuracy, retrieval recall, and interpretability compared to standard autoencoders and other KAN variants. These findings position KAE as a practical tool for high-dimensional data analysis, paving the way for more robust performance in representation learning. The code is available at `https://anonymous.4open.science/r/KAE/`.

## 1 Introduction

Autoencoders (Berahmand et al., 2024) are a fundamental component of modern deep learning, serving as powerful tools for unsupervised representation learning. By compressing input data into a lower-dimensional latent space and subsequently reconstructing it, autoencoders facilitate a wide range of applications, including dimensionality reduction (Wang et al., 2016; Lin et al., 2020), image classification (Zhou et al., 2019; 2023), and data denoising (Gondara, 2016; Cui & Zdeborová, 2024). Their ability to learn meaningful representations from unlabelled data has positioned them as essential in various AI domains, from computer vision (Mishra et al., 2018; Takeishi & Kalousis, 2021) to natural language processing (Shen et al., 2020; Kim et al., 2021), significantly enhancing the performance of downstream tasks.

Traditional autoencoders typically leverage multi-layer perceptrons (MLPs), characterized by fully connected layers. In a conventional autoencoder, each layer computes its output as $y = \sigma(Wx + b)$, where $\sigma$ denotes a fixed activation function, $W$ is the learnable weight matrix, $x$ represents the input, and $b$ is the bias vector. The optimization of $W$ is traditionally performed through black-box AI systems, limiting the adaptability of the activation function to the data's underlying structure.

Recent advancements in alternative architectures, such as Kolmogorov-Arnold Networks (KANs) (Liu et al., 2024b;a), offer a compelling pathway for enhancing autoencoder performance. Unlike MLPs, KANs redefine the output as $y = f(x)$, where $f$ is a learnable activation function. This shift enables the network to adaptively learn more complex representations, moving beyond the limitations of fixed functions and linear weights used in MLPs.

In this paper, we propose the Kolmogorov-Arnold Auto-Encoder (KAE), which integrates the strengths of KANs into the autoencoder framework to create a more robust model for representation learning. However, merely substituting MLP layers with KAN layers utilizing B-spline functions may result in only marginal gains or even a decline in performance. The efficacy of KANs heavily depends on the specific type of function employed within the KAN layer. To address this chal-

lenge, we introduce a well-defined KAE architecture that utilizes learnable polynomial functions, demonstrating promising improvements in representation learning.

Our contributions are as follows:

- We introduce the Kolmogorov-Arnold Representation Theorem into the design of autoencoders, providing a theoretical basis for improving autoencoder performance.
- We investigate the role of activation functions in the KAN layers, identifying polynomial functions as a suitable choice for enhancing autoencoder performance.
- We demonstrate the superiority of the Kolmogorov-Arnold Auto-Encoder (KAE) through extensive experimental validation, showing improvements in both reconstruction quality and downstream application performance.

The remainder of the paper is organized as follows. Section 2 reviews related work. Section 3 details the proposed KAE model, and Section 4 provides a thorough empirical evaluation. Finally, Section 5 concludes the paper.

## 2 RELATED WORK

### 2.1 AUTOENCODERS (AES)

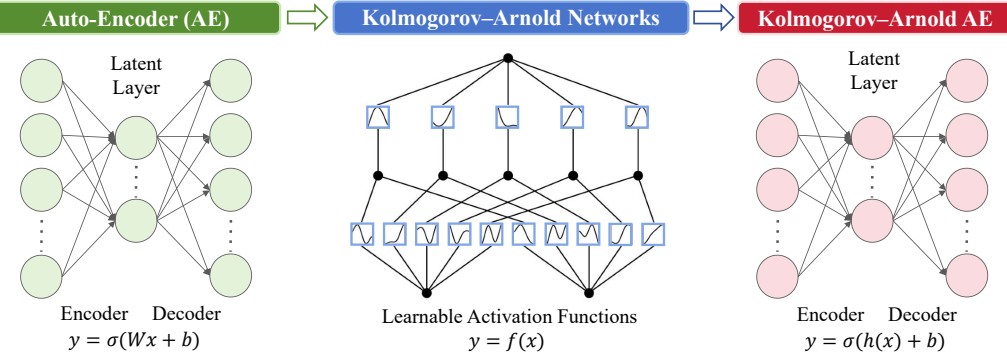

Figure 1: **Model Comparison of AE, KAN, and KAE.**

Autoencoders, as a form of unsupervised learning, aim to learn a compressed representation of input data while minimizing reconstruction error (Pinaya et al., 2020). Traditionally, autoencoders rely on MLPs to achieve this task, where each layer applies a fixed non-linearity. However, this often limits the ability of the network to capture more complex structures in the data.

Recent works have explored various autoencoder architectures, including Variational Auto-Encoders (VAEs) (Kingma et al., 2019; Skopek et al., 2020) and Denoising Auto-Encoders (DAEs) (Savinov et al., 2022; Wu et al., 2023), which introduce stochasticity and robustness to noise, respectively. Nevertheless, most of these architectures still rely on fixed activation functions, which limits their flexibility in representing complex functions.

As shown in Fig. 1, an autoencoder consists of two main parts:

- **Encoder**: This part compresses the input into a latent-space representation.
- **Decoder**: This part reconstructs the input data from the latent-space representation.

A common application of autoencoders is representation learning, where latent representations are learned and can be used for tasks such as image classification (Zhou et al., 2023), text classification (Xu & Tan, 2019), cross-modal learning (Dong et al., 2023), and other applications, particularly beneficial for high-dimensional data analysis. Another key feature of autoencoders is their ability to reduce noise in data (Bajaj et al., 2020). By inputting noisy data, the pre-trained autoencoder can remove the noise, producing clean outputs, which makes it a powerful tool for data denoising.

## 2.2 Kolmogorov-Arnold Networks (KANs)

Kolmogorov-Arnold Networks (KANs) (Liu et al., 2024b;a) provide a more flexible way to model non-linear relationships by using learnable activation functions rather than fixed ones. Inspired by the Kolmogorov-Arnold representation theorem (Kolmogorov, 1961; Braun & Griebel, 2009), KANs approximate any continuous multivariate function using a sum of univariate functions, providing a theoretically grounded approach to function approximation.

In the context of neural networks, KANs allow each layer to learn its own activation function, making them more adaptable to highly non-linear data. KANs have been shown to outperform MLPs in various applications (Xu et al., 2024; Bozorgasl & Chen, 2024), particularly those requiring complex, non-linear transformations of input data.

### 2.2.1 Kolmogorov-Arnold Representation Theorem

The Kolmogorov-Arnold representation theorem is a fundamental result in mathematics, particularly in functional analysis and multivariate approximation theory. It provides key insights into representing multivariate continuous functions. Theorem 1 asserts that any continuous function of $d$ variables can be represented as a finite sum of continuous univariate functions along with an additional continuous function.

**Theorem 1** (**Kolmogorov-Arnold Representation Theorem**). *For any smooth function* $f$ : $[0, 1]^d \to \mathbb{R}$, *there exist continuous functions* $\phi_{k,j} : [0, 1] \to \mathbb{R}$ *and* $\Phi_k : \mathbb{R} \to \mathbb{R}$ *such that:*

$$f(x_1, x_2, \ldots, x_d) = \sum_{k=1}^{2d+1} \Phi_k \left( \sum_{j=1}^{d} \phi_{k,j}(x_j) \right). \tag{1}$$

This remarkable theorem involves the use of two primary types of functions: **inner functions** $\{\phi_{k,j}\}$ and **outer functions** $\{\Phi_k\}$, which motivates the design of the KAN network.

### 2.2.2 KAN Network and Its Applications

Inspired by Theorem 1, Liu et al. (2024b) proposed a novel KAN layer with $d_{\text{in}}$-dimensional inputs and $d_{\text{out}}$-dimensional outputs, defined as a matrix of 1D functions:

$$\mathbf{\Phi} := \{\phi_{k,j}\}, \quad j = 1, 2, \cdots, d_{\text{in}}, \quad k = 1, 2, \cdots, d_{\text{out}},$$

where each function $\phi_{k,j}$ is a learnable univariate function. A general KAN network is formed by stacking $L$ KAN layers. Given an input vector $x \in \mathbb{R}^d$, the output of KAN is

$$\text{KAN}(x) := (\Phi_{L-1} \circ \Phi_{L-2} \circ \cdots \circ \Phi_1 \circ \Phi_0) \circ x, \tag{2}$$

where the Kolmogorov-Arnold representation in Eq. (1) can be viewed as a composition of two KAN layers: $\Phi_0$ contains $(2d + 1) \cdot d$ functions, and $\Phi_1$ contains $1 \cdot (2d + 1)$ functions.

Recently, KAN networks have been applied to various domains, including scientific discovery (Liu et al., 2024b;a), image segmentation (Li et al., 2024), image classification (Cheon, 2024), text classification (Imran & Ishmam, 2024), collaborative filtering (Xu et al., 2024), and others. KAN-based architectures, such as the Kolmogorov-Arnold Transformer (Yang & Wang, 2024) and UNet-KAN (U-KAN) (Li et al., 2024), have also gained significant attention.

## 3 Kolmogorov−Arnold Auto-Encoder

### 3.1 KANs in Autoencoders

While KANs offer a promising way to enhance neural network architectures, their direct application to autoencoders presents challenges. Specifically, the complexity introduced by the learnable activation functions can lead to overfitting or suboptimal performance if not carefully managed. In this work, we propose a new autoencoder architecture that integrates KANs while addressing these challenges through the use of polynomial-based activation functions, which we show to be more stable and effective for this task.

## 3.2 OVERALL ARCHITECTURE

The key idea behind the Kolmogorov-Arnold Auto-Encoder (KAE) is to replace the MLP layers (fixed activation functions) in traditional autoencoders with KAN layers (learnable functions), as inspired by the Kolmogorov-Arnold representation theorem.

In a traditional autoencoder, the MLP consists of fully connected layers that apply a fixed activation function, such as ReLU or sigmoid, after a linear transformation. Given an input vector $x \in \mathbb{R}^d$, the encoder compresses the data into a lower-dimensional latent representation $z \in \mathbb{R}^k$, and the decoder reconstructs $z$ back into the original space. This transformation is typically expressed as:

$$z = \sigma(Wx + b),$$

where $W$ is the weight matrix, $b$ is the bias, and $\sigma$ is a fixed activation function.

In KAE, we replace the MLP layers with KAN layers, which use learnable activation functions as defined by the Kolmogorov-Arnold representation theorem. Instead of a fixed $\sigma$, each KAN layer dynamically learns its own activation function, enabling the model to capture complex, non-linear patterns. The encoder applies a series of KAN layers to map the input to the latent space:

$$z = f(x) := (\Phi_{L-1} \circ \Phi_{L-2} \circ \cdots \circ \Phi_1 \circ \Phi_0) \circ x,$$

where each $\Phi_i$ represents a KAN layer composed of learnable univariate functions.

Similarly, the decoder ($y = g(z)$) uses KAN layers in reverse to reconstruct the data, providing greater flexibility and accuracy compared to standard MLP-based decoders. This architecture enables KAE to capture complex, non-linear relationships, enhancing the quality of the learned representations. As with standard autoencoders, we use the mean squared error (MSE) between the original and reconstructed data as the training loss.

## 3.3 POLYNOMIAL ACTIVATION FUNCTION

The autoencoder's primary task is to ensure that the encoding and decoding functions are inverses of each other, i.e., $x \approx g(f(x))$ for all $x \in \mathbb{R}^d$. In a traditional autoencoder, the layers are fully connected, and this inversion property is straightforward to achieve. However, when using KAN layers, the choice of functions $f$ and $g$ becomes more nuanced. Naïvely replacing the MLP layer with a KAN layer does not produce satisfactory results, as shown in our evaluation in Section 4.

To ensure consistency between the encoder and decoder, we carefully design the $f(x)$ and $g(z)$ to be invertible, using polynomial approximations that are both smooth and differentiable. For a KAE layer with $d_{\text{in}}$-dimensional inputs and $d_{\text{out}}$-dimensional outputs, the output of $x \in \mathbb{R}^{d_{\text{in}}}$ is defined as:

$$\text{KAE}(x) := \sigma(h(x) + b) = \sigma((c_0 1_{d_{\text{in}}} + c_1 x + c_2 x^2 + \cdots + c_p x^p) + b), \tag{3}$$

where $1_{d_{\text{in}}}$ is an all-ones vector and $p$ is the order of the polynomials, treated as a hyperparameter in KAE. To maintain consistency with the MLP-based AE, we use the structure $\sigma(\cdot + b)$, replacing the $Wx$ term with $h(x)$, resulting in $\sigma(h(x) + b)$ with the following advantages.

- For polynomial orders up to four, an inverse function exists, ensuring the required inversion between the encoder and decoder.
- Compared to the linear transformation $Wx$ in MLPs, polynomial functions $h(x)$ introduce higher-order non-linear terms such as $x^p$, enabling the model to capture more complex, non-linear relationships in the data.
- The polynomial function includes a constant matrix $c_0 \in \mathbb{R}^{d_{\text{out}} \times d_{\text{in}}}$, providing more flexibility than the traditional bias term $b \in \mathbb{R}^{d_{\text{out}}}$, allowing the model to better adapt to shifts and variations in the data.

Notably, in the original KAN (Liu et al., 2024b), the learnable function $h(x)$ is set as a B-spline function, while in FourierKAN (Xu et al., 2024) it is set as a Fourier function, and in WavKAN (Bozorgasl & Chen, 2024) it is set as a Wavelet function. Comparatively, our empirical validation shows that quadratic and cubic polynomial functions (with $p = 2, 3$) offer a more effective balance between flexibility and stability, improving the model's ability to reconstruct data while maintaining the inversion property between the encoder and decoder.

## 4 EVALUATION

### 4.1 EXPERIMENTAL SETUP

To assess the effectiveness of the proposed Kolmogorov-Arnold Auto-Encoder (**KAE**), we conducted a series of experiments comparing its performance against several baseline models: the standard autoencoder (**AE**) (Pinaya et al., 2020), the Kolmogorov-Arnold Network (**KAN**) (Liu et al., 2024b), and its variants, including **FourierKAN** (Xu et al., 2024) and **WavKAN** (Bozorgasl & Chen, 2024). The implementation details and hyperparameters for each model are provided in Appendix A. Our evaluation was designed to address the following key objectives:

- **Bidirectional representation learning**: Evaluating the model's ability to effectively compress and decompress data by comparing the reconstructed outputs to the original inputs;
- **Quality of latent representations**: Assessing the learned representations in downstream applications such as similarity search, image classification, and image denoising;
- **Model capacity and interpretability**: Analyzing the interpretability of the trained model by examining the learned function coefficients within the autoencoder architecture.

We employed several well-known image datasets for our evaluations, as summarized in Table 1.

Table 1: **Statistics of the Image Datasets Used in Our Work.**

| Dataset | Image Type | Image Size | #Classes | #Training | #Test |
|---|---|---|---|---|---|
| MNIST (LeCun et al., 1998) | Grayscale handwritten digits | 28×28 | 10 | 60,000 | 10,000 |
| FashionMNIST (Xiao et al., 2017) | Grayscale images of clothing | 28×28 | 10 | 60,000 | 10,000 |
| CIFAR10 (Krizhevsky & Hinton, 2009) | RGB natural images | 32×32 | 10 | 50,000 | 10,000 |
| CIFAR100 (Krizhevsky & Hinton, 2009) | RGB natural images | 32×32 | 100 | 50,000 | 10,000 |

Each experiment was repeated ten times with random seeds from 2,024 to 2,033, and the average results with standard deviation were reported. Models were trained using the Adam optimizer (Kingma, 2014), exploring four configurations of learning rate (1e-4 or 1e-5) and weight decay (1e-4 or 1e-5), with a batch size of 256 for 10 epochs. The best-performing configuration was reported.

All experiments were conducted using Python (version 3.10) and PyTorch 2.4 as the deep learning framework. Computations were performed on a ThinkStation equipped with an Intel i7-12700 CPU (2.1 GHz), 32GB of RAM, and an NVIDIA TITAN V GPU with 12GB of GPU memory.

### 4.2 RECONSTRUCTION QUALITY

Autoencoders perform the bidirectional representation learning by compressing input data into a latent space (encoding) and then reconstructing it back (decoding). To assess how well different models perform this task, we compared their reconstruction error using the mean squared error (MSE) between the original input and the reconstructed output on the test set.

For all models, we employed a shallow architecture consisting of three layers: $d_{\text{original}}$-$d_{\text{latent}}$-$d_{\text{original}}$, where $d_{\text{latent}}$ represents the dimension of the compressed latent space. After training each model, a previously unseen test input $x$ was passed through the network to obtain the latent representation $y$ and the reconstructed output $z$. The reconstruction error was then calculated as the MSE, defined by $\|x - z\|^2$, and averaged across all test data batches.

Table 2 demonstrates the superiority of KAE compared to AE and KAN variants in terms of reconstruction error. The results highlight several key observations:

- **KAE vs AE**: KAE consistently delivers significantly lower reconstruction errors than AE in all settings, with the error reduction clearly highlighted in the last row of Table 2.
- **KAE vs KAN**: The performance of KAN variants is highly dependent on the choice of activation function. The polynomial function used in our KAE model outperforms the B-spline function in KAN, the Fourier function in FourierKAN, and the wavelet function in WavKAN for this reconstruction task. Additionally, higher-order polynomial functions (i.e., $p = 3$) in KAE lead to better reconstruction performance.

Table 2: **Reconstruction Error Comparison Across Datasets for Different Latent Dimensions.** The best results are in **bold** and the second best are underlined. The last row shows the improvement of KAE models over standard autoencoders (AE).

| Dataset | MNIST | | FashionMNIST | | CIFAR10 | | CIFAR100 | |
|---|---|---|---|---|---|---|---|---|
| Dimension | 16 | 32 | 16 | 32 | 16 | 32 | 16 | 32 |
| AE | $0.056_{\pm0.002}$ | $0.043_{\pm0.001}$ | $0.045_{\pm0.002}$ | $0.034_{\pm0.001}$ | $0.034_{\pm0.001}$ | $0.029_{\pm0.001}$ | $0.037_{\pm0.001}$ | $0.030_{\pm0.001}$ |
| KAN | $0.047_{\pm0.002}$ | $0.036_{\pm0.001}$ | $0.032_{\pm0.003}$ | $0.024_{\pm0.000}$ | $0.025_{\pm0.001}$ | $0.019_{\pm0.000}$ | $0.025_{\pm0.001}$ | $0.018_{\pm0.001}$ |
| FourierKAN | $0.042_{\pm0.003}$ | $0.031_{\pm0.003}$ | $0.031_{\pm0.001}$ | $0.024_{\pm0.001}$ | $0.031_{\pm0.002}$ | $0.023_{\pm0.001}$ | $0.029_{\pm0.001}$ | $0.022_{\pm0.001}$ |
| WavKAN | $0.175_{\pm0.002}$ | $0.161_{\pm0.002}$ | $0.099_{\pm0.001}$ | $0.089_{\pm0.000}$ | $0.035_{\pm0.001}$ | $0.025_{\pm0.000}$ | $0.036_{\pm0.001}$ | $0.026_{\pm0.000}$ |
| KAE (p=1) | $0.050_{\pm0.004}$ | $0.041_{\pm0.000}$ | $0.029_{\pm0.001}$ | $0.025_{\pm0.001}$ | $0.021_{\pm0.000}$ | $0.020_{\pm0.000}$ | $0.021_{\pm0.001}$ | $0.019_{\pm0.000}$ |
| KAE (p=2) | $\underline{0.026}_{\pm0.002}$ | $\underline{0.017}_{\pm0.001}$ | $\underline{0.020}_{\pm0.000}$ | $\underline{0.016}_{\pm0.001}$ | $\underline{0.017}_{\pm0.001}$ | $\underline{0.013}_{\pm0.000}$ | $\underline{0.017}_{\pm0.001}$ | $\underline{0.013}_{\pm0.000}$ |
| KAE (p=3) | $\mathbf{0.024}_{\pm0.001}$ | $\mathbf{0.015}_{\pm0.001}$ | $\mathbf{0.018}_{\pm0.001}$ | $\mathbf{0.015}_{\pm0.000}$ | $\mathbf{0.016}_{\pm0.001}$ | $\mathbf{0.012}_{\pm0.000}$ | $\mathbf{0.016}_{\pm0.001}$ | $\mathbf{0.012}_{\pm0.000}$ |
| Improve | 0.032 ↓ | 0.028 ↓ | 0.027 ↓ | 0.019 ↓ | 0.018 ↓ | 0.017 ↓ | 0.021 ↓ | 0.018 ↓ |

## 4.3 APPLICATIONS

To evaluate the quality of the compressed data, we applied the latent representations to several downstream tasks, including similarity search, image classification, and image denoising, demonstrating the high-quality representations learned by the proposed KAE models.

### 4.3.1 SIMILARITY SEARCH

A well-designed autoencoder, as a tool for dimensionality reduction, should preserve the distance relationships between samples from the input space to the latent space, making similarity search a suitable application for assessing this property.

For each experiment, we randomly selected a subset of 1,000 test samples as the test set. For each query, we computed the $k$ nearest neighbors in the input space as the ground truth and compared it to the $N$ nearest neighbors retrieved in the latent space. We calculated the recall, defined as the ratio of true top-$k$ results within the top-$N$ retrieved candidates, and reported it as Recall $k@N$. In our experiments, we set $k = 10$ and tested Recall $10@N$ (referred to as **Recall@$N$**) for all models. Each experiment was repeated 10 times with different random seeds, and we reported the averaged Recall with standard deviations.

Table 3 presents the Recall@10 results for various models across latent dimensions (16 and 32). Increasing the latent dimension from 16 to 32 significantly improved retrieval recall for all models, enhancing their distance-preserving properties and expressive power. Notably, the proposed KAE model consistently outperformed both the standard AE and KAN variants across all datasets and dimensions. Specifically, KAE ($p = 2$) achieved improvements of 0.243 and 0.206 in Recall over AE for MNIST with latent dimensions of 16 and 32, respectively, with similar gains observed for FashionMNIST, CIFAR10, and CIFAR100 datasets.

Table 3: **Retrieval Recall@10 Comparison Across Datasets for Different Latent Dimensions.** The best results are in **bold** and the second best are underlined. The last row shows the improvement of KAE models over standard autoencoders (AE).

| Dataset | MNIST | | FashionMNIST | | CIFAR10 | | CIFAR100 | |
|---|---|---|---|---|---|---|---|---|
| Dimension | 16 | 32 | 16 | 32 | 16 | 32 | 16 | 32 |
| AE | $0.354_{\pm0.017}$ | $0.483_{\pm0.008}$ | $0.401_{\pm0.016}$ | $0.526_{\pm0.009}$ | $0.368_{\pm0.026}$ | $0.472_{\pm0.012}$ | $0.380_{\pm0.015}$ | $0.477_{\pm0.009}$ |
| KAN | $0.457_{\pm0.015}$ | $0.552_{\pm0.016}$ | $0.495_{\pm0.023}$ | $0.578_{\pm0.004}$ | $0.377_{\pm0.015}$ | $0.496_{\pm0.007}$ | $0.391_{\pm0.009}$ | $0.512_{\pm0.007}$ |
| FourierKAN | $0.498_{\pm0.053}$ | $0.638_{\pm0.034}$ | $0.518_{\pm0.022}$ | $\underline{0.615}_{\pm0.018}$ | $0.258_{\pm0.041}$ | $0.406_{\pm0.028}$ | $0.316_{\pm0.026}$ | $0.435_{\pm0.019}$ |
| WavKAN | $0.259_{\pm0.037}$ | $0.447_{\pm0.018}$ | $0.387_{\pm0.015}$ | $0.488_{\pm0.006}$ | $0.258_{\pm0.016}$ | $0.428_{\pm0.012}$ | $0.259_{\pm0.013}$ | $0.430_{\pm0.011}$ |
| KAE (p=1) | $0.404_{\pm0.028}$ | $0.488_{\pm0.007}$ | $0.544_{\pm0.008}$ | $0.581_{\pm0.010}$ | $0.440_{\pm0.011}$ | $0.489_{\pm0.008}$ | $0.453_{\pm0.008}$ | $0.504_{\pm0.005}$ |
| KAE (p=2) | $\mathbf{0.596}_{\pm0.019}$ | $\mathbf{0.689}_{\pm0.011}$ | $\mathbf{0.607}_{\pm0.010}$ | $\mathbf{0.661}_{\pm0.007}$ | $\mathbf{0.525}_{\pm0.021}$ | $\mathbf{0.631}_{\pm0.011}$ | $\mathbf{0.544}_{\pm0.013}$ | $\mathbf{0.639}_{\pm0.007}$ |
| KAE (p=3) | $\underline{0.554}_{\pm0.013}$ | $\underline{0.659}_{\pm0.013}$ | $\underline{0.521}_{\pm0.006}$ | $0.582_{\pm0.006}$ | $\underline{0.488}_{\pm0.012}$ | $\underline{0.597}_{\pm0.010}$ | $\underline{0.493}_{\pm0.013}$ | $\underline{0.586}_{\pm0.012}$ |
| Improve | 0.242 ↑ | 0.206 ↑ | 0.206 ↑ | 0.135 ↑ | 0.157 ↑ | 0.159 ↑ | 0.164 ↑ | 0.162 ↑ |

As shown in Fig. 2, increasing the number of retrieved samples revealed that non-linear polynomial functions (e.g., KAE ($p = 2, 3$)) achieved competitive results. Especially on more complex datasets like CIFAR10 and CIFAR100, KAEs significantly outperformed FourierKAN and WavKAN, which exhibited lower recall values. This highlights the ability of KAEs to preserve the intrinsic structure of the data in the latent space, making them highly effective for similarity-based retrieval tasks.

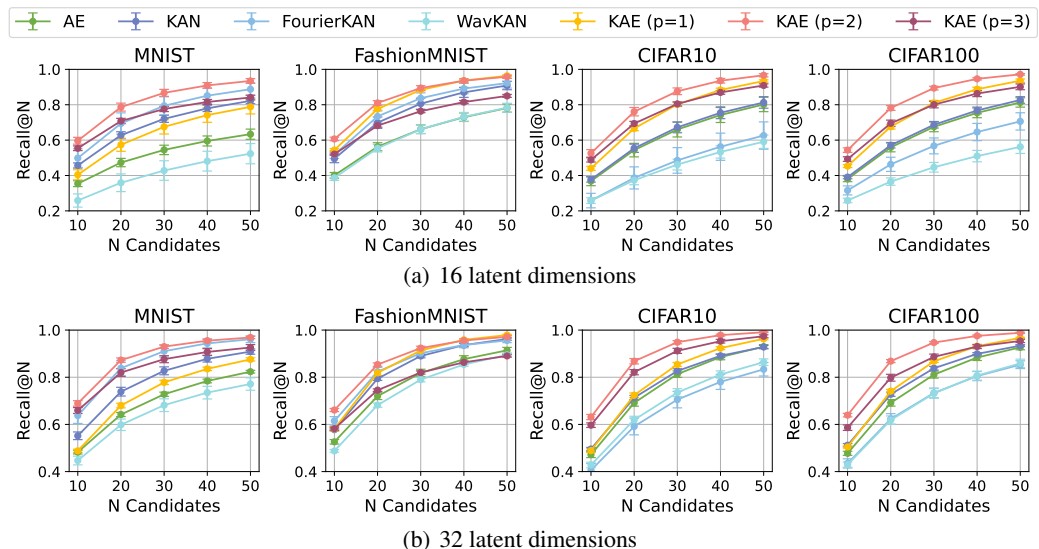

(a) 16 latent dimensions

(b) 32 latent dimensions

Figure 2: **Recall@$N$ of Similarity Search Across Datasets for Different Latent Dimensions**.

### 4.3.2 IMAGE CLASSIFICATION

We further applied the learned latent representations to image classification using a nearest neighbor classifier. For each of the 10,000 test samples, the predicted label was assigned based on the nearest sample in the latent space with the smallest Euclidean distance. We then compared the predicted labels with the ground truth, averaging the classification accuracy across all test samples.

Table 4 shows that KAE models, particularly with the non-linear polynomial function ($p = 3$), achieved the highest classification accuracy across all datasets. This contrasts with similarity search, where $p = 2$ performed best. The difference in performance may due to the nature of each task:

- In similarity search, KAE ($p = 2$) better preserves the distance relationships between neighbors, which is crucial for retrieving similar samples.

- In image classification, KAE ($p = 3$) captures more complex, non-linear relationships, enabling better discrimination between class boundaries and improving accuracy.

Table 4: **Classification Accuracy Comparison Across Datasets for Different Latent Dimensions.** The best results are in **bold** and the second best are underlined. The last row shows the improvement of KAE models over standard autoencoders (AE).

| Dataset | MNIST | | FashionMNIST | | CIFAR10 | | CIFAR100 | |
|---|---|---|---|---|---|---|---|---|
| Dimension | 16 | 32 | 16 | 32 | 16 | 32 | 16 | 32 |
| AE | $0.853_{\pm 0.014}$ | $0.916_{\pm 0.007}$ | $0.737_{\pm 0.007}$ | $0.781_{\pm 0.004}$ | $0.243_{\pm 0.010}$ | $0.283_{\pm 0.006}$ | $0.076_{\pm 0.005}$ | $0.101_{\pm 0.003}$ |
| KAN | $0.883_{\pm 0.012}$ | $0.931_{\pm 0.006}$ | $0.752_{\pm 0.007}$ | $0.786_{\pm 0.002}$ | $0.244_{\pm 0.006}$ | $0.290_{\pm 0.004}$ | $0.080_{\pm 0.003}$ | $0.110_{\pm 0.003}$ |
| FourierKAN | $0.859_{\pm 0.047}$ | $0.947_{\pm 0.012}$ | $0.735_{\pm 0.008}$ | $0.795_{\pm 0.007}$ | $0.164_{\pm 0.016}$ | $0.246_{\pm 0.017}$ | $0.040_{\pm 0.007}$ | $0.085_{\pm 0.009}$ |
| WavKAN | $0.649_{\pm 0.056}$ | $0.887_{\pm 0.012}$ | $0.679_{\pm 0.009}$ | $0.751_{\pm 0.004}$ | $0.189_{\pm 0.008}$ | $0.272_{\pm 0.005}$ | $0.043_{\pm 0.003}$ | $0.096_{\pm 0.003}$ |
| KAE (p=1) | $0.802_{\pm 0.013}$ | $0.868_{\pm 0.008}$ | $0.751_{\pm 0.005}$ | $0.773_{\pm 0.005}$ | $0.262_{\pm 0.006}$ | $0.282_{\pm 0.003}$ | $0.087_{\pm 0.003}$ | $0.104_{\pm 0.002}$ |
| KAE (p=2) | $\underline{0.929}_{\pm 0.011}$ | $\underline{0.963}_{\pm 0.002}$ | $\underline{0.801}_{\pm 0.003}$ | $\underline{0.824}_{\pm 0.003}$ | $\underline{0.304}_{\pm 0.010}$ | $\underline{0.338}_{\pm 0.004}$ | $\underline{0.124}_{\pm 0.006}$ | $\underline{0.148}_{\pm 0.002}$ |
| KAE (p=3) | $\mathbf{0.940}_{\pm 0.005}$ | $\mathbf{0.964}_{\pm 0.002}$ | $\mathbf{0.805}_{\pm 0.004}$ | $\mathbf{0.826}_{\pm 0.002}$ | $\mathbf{0.315}_{\pm 0.009}$ | $\mathbf{0.354}_{\pm 0.004}$ | $\mathbf{0.131}_{\pm 0.005}$ | $\mathbf{0.154}_{\pm 0.003}$ |
| Improve | 0.087 ↑ | 0.048 ↑ | 0.068 ↑ | 0.045 ↑ | 0.072 ↑ | 0.071 ↑ | 0.055 ↑ | 0.053 ↑ |

### 4.3.3 IMAGE DENOISING

Image denoising is a natural extension of autoencoder applications, used to evaluate the robustness of learned models. By adding noise to the input images, we can assess the model's ability to remove noise through compression and reconstruction, using the mean squared error (MSE) between the denoised and original clean images as denoising error. We applied two common types of noise:

- **Gaussian noise**: Added Gaussian noise $\mathcal{N}(0, 0.1^2)$ to the clean images.
- **Salt-and-Pepper noise**: Randomly set pixels to 0 or 1 with a probability of 0.05.

Results in Table 5 show that both $p = 2$ and $p = 3$ of KAE consistently achieved the lowest denoising errors across all datasets, mirroring their strong performance in reconstruction. This indicates that the KAE models not only excel in image reconstruction but are also highly effective in removing noise, demonstrating their robustness in preserving the underlying structure of data. The consistent performance across different types of noise further highlights the KAE models' ability to generalize well to varying noise conditions, making them superior to both the standard AE and KAN models.

Table 5: **Denoising Error Comparison Across Datasets for Different Latent Dimensions.** The best results are in **bold** and the second best are underlined. The last row shows the improvement of KAE models over standard autoencoders (AE).

| Dataset | MNIST | | FashionMNIST | | CIFAR10 | | CIFAR100 | |
|---|---|---|---|---|---|---|---|---|
| Dimension | 16 | 32 | 16 | 32 | 16 | 32 | 16 | 32 |
| | | | | *I. Gaussian Noise* | | | | |
| AE | $0.065_{\pm 0.002}$ | $0.053_{\pm 0.001}$ | $0.056_{\pm 0.002}$ | $0.044_{\pm 0.001}$ | $0.044_{\pm 0.001}$ | $0.038_{\pm 0.001}$ | $0.047_{\pm 0.001}$ | $0.040_{\pm 0.001}$ |
| KAN | $0.058_{\pm 0.002}$ | $0.046_{\pm 0.001}$ | $0.043_{\pm 0.003}$ | $0.034_{\pm 0.000}$ | $0.035_{\pm 0.001}$ | $0.029_{\pm 0.000}$ | $0.034_{\pm 0.001}$ | $0.028_{\pm 0.001}$ |
| FourierKAN | $0.063_{\pm 0.002}$ | $0.054_{\pm 0.001}$ | $0.049_{\pm 0.001}$ | $0.041_{\pm 0.001}$ | $0.048_{\pm 0.002}$ | $0.041_{\pm 0.001}$ | $0.047_{\pm 0.001}$ | $0.040_{\pm 0.001}$ |
| WavKAN | $0.188_{\pm 0.003}$ | $0.174_{\pm 0.004}$ | $0.115_{\pm 0.002}$ | $0.105_{\pm 0.001}$ | $0.045_{\pm 0.001}$ | $0.035_{\pm 0.000}$ | $0.046_{\pm 0.001}$ | $0.037_{\pm 0.000}$ |
| KAE (p=1) | $0.058_{\pm 0.004}$ | $0.051_{\pm 0.000}$ | $0.038_{\pm 0.001}$ | $0.035_{\pm 0.001}$ | $0.031_{\pm 0.000}$ | $0.030_{\pm 0.000}$ | $0.031_{\pm 0.000}$ | $0.029_{\pm 0.000}$ |
| KAE (p=2) | $0.038_{\pm 0.002}$ | $0.027_{\pm 0.001}$ | $0.030_{\pm 0.000}$ | $0.026_{\pm 0.001}$ | **$0.027_{\pm 0.001}$** | $0.023_{\pm 0.000}$ | **$0.026_{\pm 0.001}$** | **$0.022_{\pm 0.000}$** |
| KAE (p=3) | **$0.034_{\pm 0.001}$** | **$0.025_{\pm 0.001}$** | **$0.029_{\pm 0.001}$** | **$0.025_{\pm 0.000}$** | **$0.027_{\pm 0.001}$** | **$0.022_{\pm 0.000}$** | **$0.026_{\pm 0.001}$** | **$0.022_{\pm 0.000}$** |
| Improve | 0.031 ↓ | 0.028 ↓ | 0.027 ↓ | 0.019 ↓ | 0.017 ↓ | 0.016 ↓ | 0.021 ↓ | 0.018 ↓ |
| | | | | *II. Salt-and-Pepper Noise* | | | | |
| AE | $0.092_{\pm 0.002}$ | $0.082_{\pm 0.001}$ | $0.078_{\pm 0.001}$ | $0.069_{\pm 0.001}$ | $0.058_{\pm 0.001}$ | $0.053_{\pm 0.001}$ | $0.061_{\pm 0.001}$ | $0.055_{\pm 0.001}$ |
| KAN | $0.086_{\pm 0.002}$ | $0.075_{\pm 0.001}$ | $0.067_{\pm 0.002}$ | $0.060_{\pm 0.000}$ | $0.050_{\pm 0.001}$ | $0.045_{\pm 0.000}$ | $0.050_{\pm 0.001}$ | $0.045_{\pm 0.000}$ |
| FourierKAN | $0.087_{\pm 0.001}$ | $0.080_{\pm 0.002}$ | $0.071_{\pm 0.001}$ | $0.065_{\pm 0.001}$ | $0.065_{\pm 0.001}$ | $0.059_{\pm 0.001}$ | $0.064_{\pm 0.001}$ | $0.059_{\pm 0.001}$ |
| WavKAN | $0.207_{\pm 0.006}$ | $0.199_{\pm 0.013}$ | $0.139_{\pm 0.005}$ | $0.127_{\pm 0.003}$ | $0.059_{\pm 0.001}$ | $0.051_{\pm 0.000}$ | $0.061_{\pm 0.001}$ | $0.052_{\pm 0.000}$ |
| KAE (p=1) | $0.085_{\pm 0.003}$ | $0.080_{\pm 0.001}$ | $0.063_{\pm 0.001}$ | $0.061_{\pm 0.001}$ | $0.047_{\pm 0.000}$ | $0.046_{\pm 0.000}$ | $0.048_{\pm 0.000}$ | $0.046_{\pm 0.000}$ |
| KAE (p=2) | $0.070_{\pm 0.002}$ | **$0.061_{\pm 0.001}$** | **$0.057_{\pm 0.000}$** | $0.054_{\pm 0.000}$ | **$0.044_{\pm 0.001}$** | **$0.040_{\pm 0.000}$** | **$0.044_{\pm 0.000}$** | **$0.040_{\pm 0.000}$** |
| KAE (p=3) | **$0.068_{\pm 0.001}$** | **$0.061_{\pm 0.001}$** | **$0.057_{\pm 0.001}$** | **$0.053_{\pm 0.000}$** | **$0.044_{\pm 0.001}$** | **$0.040_{\pm 0.000}$** | **$0.044_{\pm 0.000}$** | **$0.040_{\pm 0.000}$** |
| Improve | 0.024 ↓ | 0.021 ↓ | 0.021 ↓ | 0.016 ↓ | 0.014 ↓ | 0.013 ↓ | 0.017 ↓ | 0.015 ↓ |

### 4.4 PERFORMANCE ANALYSIS

**Convergence Analysis.** We measured the test loss as the average MSE between the reconstructed and original data in the test set. Fig. 3 illustrates the faster convergence of our KAE models with $p = 2$ or 3, which converge within approximately 10 epochs and achieve the lowest test loss. In contrast, other models, particularly WavKAN and AE, struggle to converge even after 50 epochs.

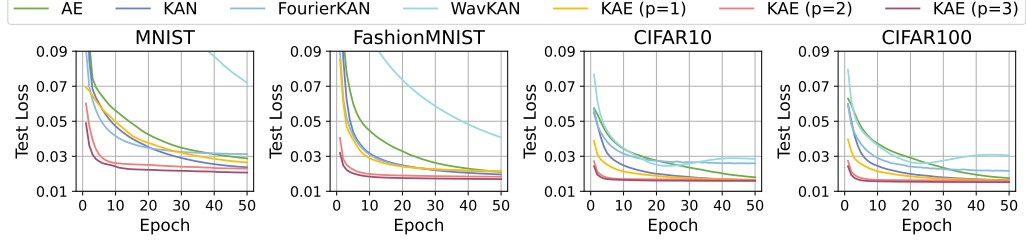

Figure 3: **Convergence Analysis of Test Loss Across Datasets for 16 Latent Dimensions**.

**Model Capacity Analysis.** As shown in Fig. 4, our KAE models strike a balance between efficiency and accuracy, with the following key observations:

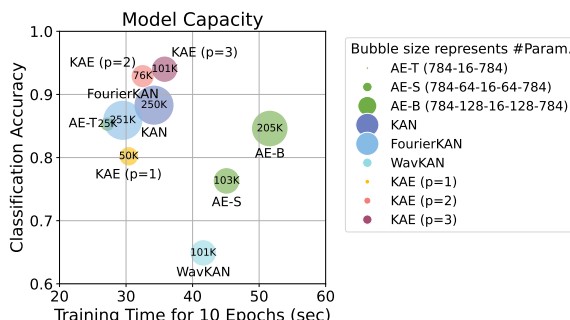

- **Training Efficiency**: While KAE models are not the fastest, they complete training in 30-36 seconds, only marginally slower than the fastest models, and significantly faster than two models with similar parameter counts, i.e., WavKAN and AE-S.

- **Classification Accuracy**: KAE models with $p = 2, 3$ use much fewer parameters (75-101K) while achieving higher accuracy compared to KAN (250K) and FourierKAN (251K).

Figure 4: **Model Capacity Analysis on the MNIST Dataset with 16 Latent Dimensions.** Bubble size represents the number of learnable parameters. For AE models, T = Tiny, S = Small, and B = Base.

- **Model Parameters**: When comparing models with similar parameter counts, KAE outperforms both WavKAN and AE-S in terms of training speed and performance. Even when the parameters of AE models are doubled, KAE still surpasses AE-B.

### 4.5 INTERPRETABILITY

Our KAE models use a polynomial activation function, learning the coefficients of $f(x) = c_0 + c_1 x + c_2 x^2 + c_3 x^3$ for $p = 3$. In the MNIST dataset with 16 latent dimensions, the input $x$ ranges from 0 to 255, making the highest order term $c_3 x^3$ dominant. Each latent node learns two 784-dimensional coefficient vectors for $c_3$, one for input features (encoder) and one for output features (decoder). Thus,

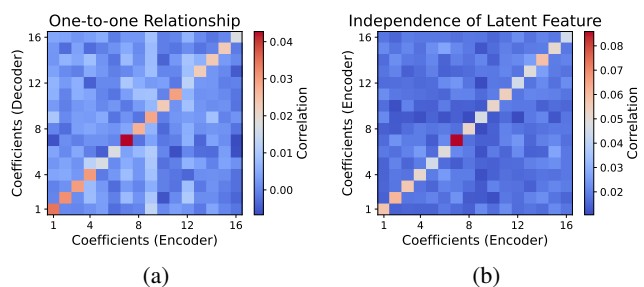

Figure 5: **Interpretability on the MNIST with $d_{\text{latent}} = 16$.**

the encoder has 16 coefficient vectors $C_E \in \mathbb{R}^{16 \times 784}$, and the decoder has $C_D \in \mathbb{R}^{16 \times 784}$.

We analyzed the relationships between these vectors. Fig. 5(a) shows $C_E C_D^\top \in \mathbb{R}^{16 \times 16}$, revealing a one-to-one correspondence between the encoder and decoder, consistent with effective compression and reconstruction. Fig. 5(b) shows $C_E C_E^\top$, demonstrating the independence of the coefficient vectors in the encoder, indicating that all latent features are meaningful and independent.

## 5 CONCLUSION

In this paper, we introduced the Kolmogorov-Arnold Auto-Encoder (KAE), which integrates the Kolmogorov-Arnold Network (KAN) with autoencoders (AEs) to create a more powerful and flexible framework for representation learning. By incorporating KAN's learnable polynomial activation functions into the AE structure, KAE effectively captures complex, non-linear relationships in the data, outperforming standard AEs. Our experiments on benchmark datasets highlight KAE's superiority in reconstruction quality and downstream applications such as similarity search, image classification, and image denoising. Our analysis further demonstrates KAE's greater capacity and interpretability, especially through its learned polynomial activation functions.

Looking ahead, future work will focus on scaling KAE to deeper architectures to unlock its potential in more complex and high-dimensional tasks. We will also explore different activation functions to further enhance the model's flexibility and performance. Additionally, applying KAE to more challenging real-world applications will provide deeper insights into its robustness and adaptability.

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

## A  APPENDIX

The implementation details and hyperparameters are listed as follows.

- **KAN**: The grid size was set to 5, with a cubic spline (order 3) as the basis for polynomial functions. The input grid range was $[-1, 1]$ for each dimension, chosen to match the training data distribution and cover the primary input space.
- **FourierKAN**: The grid size was also set to 5 for the FourierKAN model.
- **WavKAN**: We used Mexican hat wavelets for feature extraction, selected for their ability to capture local features and handle oscillatory behavior effectively.
- **KAE**: The polynomial order $p$ for the KAE model was set to 1, 2, and 3.

