# OpenReview forum: "KAE: Kolmogorov-Arnold Auto-Encoder for Representation Learning"
_ICLR.cc/2025/Conference — ICLR 2025 Conference Withdrawn Submission_

### Official Review · Reviewer_v4pi · 2024-10-18

**Soundness:** 2
**Presentation:** 3
**Contribution:** 2
**Rating:** 3
**Confidence:** 4

**Summary:**

This paper proposes using Kolmogorov Arnold networks for autoencoders instead of standard MLPs. To improve their performance they tweak the regular KAN architecture by changing the type of learnable activations and test this modified architecture on three main tasks - reconstruction, downstream classification, and denoising - showing that their architecture outperforms regular autoencoders on all three. Moreover KAE’s convergence speed, runtime, and parameter efficiency are compared to regular autoencoders.

**Strengths:**

**Scientific quality:**

The experiments are simple and extensive. I like how in the similarity experiment the representations are directly compared via nearest neighbours instead of using fine-tuning so their direct relevance is measured. It would be interesting to see if the gap holds when fine tuning but this is a very minor point.

Moreover, a decent comparisons suite between KAN variants is held. Although this isn’t mentioned in the paper, it looks like the majority of them outperform the autoencoder baseline.

Although unmentioned in the paper, regular KANs - without modifying their type of learnable activations - generally already seem to outperform the default autoencoders.

**Clarity/presentation:**

The paper is generally well written and gives a solid background.

Its purpose - testing an existing architecture in a new domain, learning small latent representations - is clear and the experiments are well presented.

**Novelty:**

This paper tests how to implement a new, popular architecture in a well-studied setting. Although not stated, this follows in spirit previous lines of works with other architectures, eg. transformers [1-3].

**References:**

[1] Dosovitskiy, A. (2020). An image is worth 16x16 words: Transformers for image recognition at scale. arXiv preprint arXiv:2010.11929.
[2] Peebles, W., & Xie, S. (2023). Scalable diffusion models with transformers. In Proceedings of the IEEE/CVF International Conference on Computer Vision (pp. 4195-4205).
[3] Chen, L., Lu, K., Rajeswaran, A., Lee, K., Grover, A., Laskin, M., ... & Mordatch, I. (2021). Decision transformer: Reinforcement learning via sequence modeling. Advances in neural information processing systems, 34, 15084-15097.

**Weaknesses:**

**Scientific quality:**

It’s unclear to what extent are the improvements due to model selection, especially as the kind of activation function used was especially selected for these tasks. Moreover, how this selection was done isn’t clear - was it based on the validation accuracy? The paper doesn’t elaborate and as it currently is seems to indicate that it was based on the test accuracy as that’s the only metric which is given.

It’s unclear if the MLP and KAE have identical computational resources - how many parameters do each have? How many FLOPs? This is somewhat indicated in Figure 4, based on which it seems that the models aren’t equally powerful, at least using these simple heuristic comparisons.

As the models have different sizes but the same hyperparameter setup was used for all, it’s unclear whether improvements are chiefly due to better inductive biases or these hyperparameters suiting some more than others, eg. as bigger models overfit more or smaller ones have implicit regularisation. As the models and datasets are modestly sized a small hyperparameter search over each setup would go a long way.

Based on section 4.5, is the data standardised before feeding it into the models? If not then why? This is a standard practice that could significantly affect learning.

For section 4.4, is the runtime comparison justified given the relatively small batch sizes and different number of parameters? It’s difficult to understand how it would extrapolate to larger, more realistic settings where the GPU is fully utilised. A single forward pass’ runtime with a big batch would be more indicative.

**Clarity/presentation:**

The paper has imprecise statements at time, eg. “does not produce satisfactory results” (line 193).

At times the paper uses unscientific language, eg. “this remarkable theorem” (133).

Information is sometimes overly repeated or overly detailed, eg. there’s no need to say so explicitly what an autoencoder is (lines 98-103) and KANs are in practice presented several times in similar ways (lines 46-49, 164-166, 174-176, etc.).

Lots of information can be delegated to the appendix or removed as it can be considered well known/irrelevant, eg. table 1.

**Novelty:**

Although it’s generally worthwhile using different architectures in new domains, generally there’s a clear benefit or problem that doing so would solve. For example, [1] used transformers for vision to a) get better compute scaling and b) show that large-scale data is more important than inductive biases. While some potential deeper improvements are stated here, eg. “a more robust model for representation learning” (line 51), “providing a theoretical basis for improving AE performance” (59), “KANs have been shown to outperform MLPs in various applications (Xu et al., 2024; Bozorgasl & Chen, 2024), particularly those requiring complex, non-linear transformations of input data” (116-118), etc., these seem speculative or insufficiently rigorous as it’s unclear how the paper shows these points.

**Questions:**

**Scientific quality:**

Based on section 3.3 lines 194-196 it looks like it’s much more work to get the KAE to work relative to the MLP, no? How exactly are f,g carefully designed? This is unclear.

In 4.3.1, lines 296-302, why are input-space nearest neighbours the ground truth? There’s many examples when that doesn’t hold, eg. pages 11-12 in https://ai.stanford.edu/~quocle/tutorial1.pdf . Is this a standard in other works? It seems quite significant. Using either a dataset with explicit ground truths or some other labelling where it’s more intuitive why some are closer to the other, eg. using latent representations of a powerful pretrained model, would be better.

As several KAE models are tested, implicitly doing model selection, why not compare different MLP activations? Especially common semi-learnable ones like various gated linear units.

Lines 96-99 - are “regular” MLP AEs not universal? How do the fixed activations limit their flexibility?

Why are only relatively shallow, MLP-analogous networks used? Why not use bigger, more realistic models? No need to use these throughout but at least one more realistic example would be interesting.

**Clarity/presentation:**

The citations in lines 116-118 show improved performance but isn’t it being especially for “complex non-linear transformations of input-data” is basically saying it’s good for representation learning? If the paper purports to show this it’s unclear where it does so.

202-204 - is this a theorem? Is it speculative? Unclear.

207-209 - is it speculated that the constant matrix provides more flexibility or a rigorous statement? Is this demonstrated in any way? Unclear.

Lines 43-44 - aren’t we always limited by the data’s underlying structure?

Lines 113-114 - what are the problems with existing approaches for showing universality?

Lines 181-182 - what does “in reverse” mean?

Lines 359-363 - is this speculative? Is this concretely shown somewhere? If the former then it should be rephrased/removed.

4.3.3, gaussian noise case - how is the data preprocessed? What’s the typical variance? Qualitative examples of the data would help.

4.5 - assuming these results are for the KAE? Wouldn’t it be possible to make a similar plot for the regular AE? It’s generally a bit unclear what figure 5 represents, recommend expanding on that in the appendix/main paper given room.

**Minor points, suggestions, etc:**

Several points I found interesting are mentioned in the “pros” here but not explicitly in the paper, consider whether to address them.

The figure 3 plots might be clearer with a logarithmic y axis.

It would be interesting to visually see the improved reconstructions from the KANs/KAE to get a qualitative comparison.

Why is the background on VAEs, denoising AEs, etc. relevant?

In section 3.3, assuming the KAE’s encoder/decoder are both universal, how exactly would the choice of f,g be more nuanced?

It would be interesting to qualitatively compare reconstructions of latent space interpolations for different models.

Recommend citing the original AE paper.

Recommend explicitly showing the KAN structure with some basis functions in section 2.2 somewhere, as it is it’s unclear what/where are its learnable parameters.

Isn’t “Bidirectional representation learning” just regular AE dimensionality reduction and reconstruction? Recommend using more standard terms.

It would be interesting to compare to VAEs as a strong baseline you’d naively expect to be better as they solve a fundamental bottleneck (degenerate vs stochastic latent space).

**Decision:**

Although this paper has a good fundamental idea its execution requires some more work and scientific precision to be adequate, I recommend the authors to work on it a bit more and try fixing these points so the comparison is clearer.

---

### Official Review · Reviewer_2aKu · 2024-10-25

**Soundness:** 1
**Presentation:** 3
**Contribution:** 1
**Rating:** 3
**Confidence:** 4

**Summary:**

The paper introduces the Kolmogorov-Arnold Auto-Encoder (KAE) by integrating Kolmogorov-Arnold Networks (KANs) with autoencoders. By using polynomial functions as learnable activation functions, the paper claims that KAE captures complex data patterns and non-linear relationships and outperforms standard autoencoders that use fixed activation functions.

**Strengths:**

1. The proposed method outperforms baseline models with minimal architectures on a selection of simple datasets.

2. The paper is overall easy to read and understand.

**Weaknesses:**

1. The experiments in the paper are insufficient to substantiate its claims (e.g., "These findings position KAE as a ‘practical’ tool for high-dimensional data analysis" and "KAE effectively captures complex data patterns"). The architecture used is too minimal to be considered empirically significant, especially given the simplicity of the datasets used. Although the authors provide experiments on four datasets, quality is more important than quantity.

2. Using learned activation functions introduces extra computational overhead compared to standard autoencoders with the same architecture but fixed activation functions. There is no section detailing this computational overhead (e.g., GPU memory usage).

**Questions:**

In the conclusion, the authors acknowledge some of the paper’s limitations, noting that future work will focus on "scaling KAE to deeper architectures" and "applying KAE to more challenging real-world applications." As it stands, however, the paper lacks both theoretical and empirical contributions necessary for acceptance. If the authors can demonstrate KAE’s advantages using larger architectures and more complex datasets, it would represent a significant empirical contribution.

Suggestions for Improvement:
1. Add experiments with larger, more complex architectures. Moreover, it would be beneficial to incorporate KAE into widely used architectures, such as transformers, CNNs, and possibly MLP-Mixer [1].

2. Add experiments with more complex real-world datasets. To strengthen empirical evidence, experiments on meaningful real-world datasets, such as ImageNet, as used in [2], are recommended.

3. Apply KAE to popular autoencoder frameworks.
Demonstrating KAE’s compatibility (and ideally superiority) with popular autoencoder frameworks, such as variational autoencoders (VAE), MAE [2], and VQ-VAE [3], would enhance its practical relevance.


[1] Ilya Tolstikhin, Neil Houlsby, Alexander Kolesnikov, Lucas Beyer, Xiaohua Zhai, Thomas Unterthiner, Jessica Yung, Daniel Keysers, Jakob Uszkoreit, Mario Lucic, et al. Mlp-mixer: An all-mlp architecture for vision. arXiv preprint arXiv:2105.01601, 2021.

[2] Kaiming He, Xinlei Chen, Saining Xie, Yanghao Li, Piotr Dollár, and Ross Girshick. Masked
autoencoders are scalable vision learners. arXiv:2111.06377, 2021.

[3] Aäron van den Oord, Oriol Vinyals, and Koray Kavukcuoglu. Neural discrete representation learning.
CoRR, abs/1711.00937, 2017.

---

### Official Review · Reviewer_qzTq · 2024-10-28

**Soundness:** 2
**Presentation:** 3
**Contribution:** 2
**Rating:** 3
**Confidence:** 4

**Summary:**

In this paper, the author adopts the newly proposed Kolmogorov-Arnold network to the basic auto-encoder network by replacing the MLP layer with KAN. In order to improve the performance, the authors propose the polynomial kernel instead of B-spline kernel in the original KAN paper. The 2-layer KAE model is shown to outperform the shadow auto-encoder and KAN with B-spline, Fourier basis, Wavelet basis in MINST, and CIFAR datasets.

**Strengths:**

- Clarity: This paper is clearly written with well-organized structure. The presentation is concise. The theorem in the paper demonstrated that using KAN can achieve universal function approximation. The structure of experiment section is also good, with KAE showing good performance in similarity search, image classification, image denoising
- Significance: KAN as a replacement of MLP has attracted attentions this year. It is useful for other researchers to understand this new technique more.

**Weaknesses:**

- Originality:

One of the major issue of this paper is the lack of originality. The KAE model is simply replacing MLP by KAN in auto-encoder with a simple change of kernel from B-spline to polynomial.

The auto-encoder network is widely used and simple to implement while KAN is novel, it is not the contribution of this paper. It is expected that to publish in ICLR we need to either propose a novel idea that would challenge the existing belief or we should present the new system that beat the state-of-the-art.  Given that the plain two-layer auto-encoder is not the state-of-the-art in either CIFAR or MINST and the two-layer architecture is trivial, we are expecting that the authors would bring new insights from either theory or in implementation point of view to show why KAE is able to challenge MLP. Unfortunately, due to the limited choice of model architecture and experiment datasets (MINST and CIFAR is hard representative of the benchmark in the ML community), this paper  is not achieving this goal yet.


- Clarity:
      - The plot is small and the line in Figure 2,3 cannot be distinguished in grey-scale printout since the markers and line styles are the same. It is suggested to modify markers and line styles so that it can be visualized without relying on colored pdf file.
      - The interpretability result only covers the MINST, which is a toy dataset. It is not very convincing. On the other hand, it is not convincing that KAE is more interpretable than MLP-based AE. Both has poor interpretability but AE at least maps linearly locally to Euclidean space but KAE maps to feature space via high dimensional kernels, which is hard to interpret.

- Significance:
    - The key challenge for KAE or KAN framework is to convince the other researchers that it worth the restructuring of the basic building block. In high-level, KAN layer is a kernel non-parametric regression model, which is in contrast to the MLP layer which is a parametric model.
        	- The benefit of non-parametric regression is that it learns from a function space which is very high dimensional compared to original parametric space.
        	- The drawback of this framework, however, lies in the choice of kernel. Like the difficulty we faced in early 2000s, kernel machine need to select a family of kernels apriori, which is hand-crafted. KAE, KAN with B-spline, wavelet, fourier basis are all human crafted feature family. This is the same issue for authors. For your use case which is MINST and CIFAR, polynomial kernel may be good enough. But how about NLP tasks? how about ImageNet, how to choose a family of kernel? in fact, it is critical to choose a good family of kernels for this type of system to work the best. But there is no guidance on that.
     - Another issue is that why only use shadow network? is there limitation for KAN to go deep? AE can benefit strongly by going deep. This is a fundamental question for KAN, since each layer is very expressive, it seems that there is little to learn for the second KAN layer on top. This paper could demonstrate the performance gain by constructing a 6 layer + network, if possible.
    -  Finally, it is the question on whether or not the KAN suffers from vanishing and exploding gradient issue more than MLP. Kernel such as polynomial kernel is not stationary. In a sense, in a long run, the error in gradient computation will grew much faster than ReLU activation. This is a critical issue if we want to go deep.

**Questions:**

-  The plot is small and the line in Figure 2,3 cannot be distinguished in grey-scale printout since the markers and line styles are the same. It is suggested to modify markers and line styles so that it can be visualized without relying on colored pdf file.
-  The interpretability result only covers the MINST, which is a toy dataset. It is not very convincing. It is expected that at least it covers the CIFAR dataset and compared with AE
-  More experiments that compared performance of AE and KAE under deep architecture setup is necessary to understand the scalability of this framework.
-  Polynomial kernel is non-stationary, meaning it more likely suffer from vanishing and exploding gradient as compared to ReLU. How to address this issue under KAE?

---

### Official Review · Reviewer_ipFz · 2024-11-03

**Soundness:** 2
**Presentation:** 3
**Contribution:** 2
**Rating:** 3
**Confidence:** 4

**Summary:**

The paper introduces the Kolmogorov-Arnold Arnold Autoencoder, which extends the idea of Kolmogorov-Arnold networks (KANs) to auto-encoding architectures. The authors find that the choice of nonlinearity is essential for KANs in the auto-encoding context and  perform differently for KANs in the auto-encoding context and observe that lower-order polynomials perform best, potentially due to the easing inversion of the encoder.

**Strengths:**

- Well-written and clearly communicated
- Potentially interesting alternative to standard autoencoders
- Thorough ablation on the activation functions

Overall a good paper and interesting results and probably good development but the experiments really lack the evidence for the bold claims made in the abstract and introduction. The paper should be revised on that side and at the same time put evidence forward for made claims, for example, repeatedly stating the superior learning of general functions of KANS comapred to MLPs, which is simply not proved empirically nor theoretically. Further, experiments are needed with more hidden dimensionality to see the trend when increasing it whether the MLP would perform better as the current data suggests that there is a cross-over point for higher dimensionality. Also AEs are rarely learned with just 16 to 32 hidden dimensinoality making this a very one-sided comparison. Lastly, I want to question the AE architecture employed. The authors state that the architecture has three layers, this means that either the encoder or the decoder can only have 1 layer, which is not sufficient to learn actual nonlinear patterns, since it is just a single linear layer plus potential activation. This is very problematic.

**Weaknesses:**

- Claims like demonstrating "superiority of the Kolmogorov-Arnold Auto-Encoder (KAE) through extensive experimental validation" are not supported by the empirical evidence because the experiments are conducted exclusively with shallow auto-encoder baselines that are quite unrealistic and not used in practice.
- Too strong claims on the power of KANs themselves like stating that MLPs "still rely on fixed activation functions, which limits their flexibility in representing complex functions". This is neither practically nor theoretically true and is also just stated without even a reference. With enough layers and/or width, MLPs should be equally powerful.
- Insufficient experimental validation: the experiments are conducted on standard benchmarks but with extremely small architectures (3 layers) and hidden dimensionality of 16 or 32. Further, image datasets should employ a convolutional architecture instead of MLPs, at least as a point of comparison what is achievable.

Overall, I think the paper could be improved by toning down claims and improving the span of experiments. Currently, while the paper is very well written and illustrated, I find it scientifically questionable due to above mentioned claims and limited experimental validation. The paper hinges on practicality, which has not been verified by the limited scale experiments. I am open to changing my score in case these two major concerns are addressed.

**Questions:**

In line 256, it is stated that the encoder-decoder network uses a shallow 3-layer architecture with $d_\text{original} - d_\text{latent} - d_\text{original}$. This seems insufficient for an MLP-based auto-encoder because it would have either a single layer for encoding or decoding making it effectively a linear model for one of them, while this might be fine for a KAN due to the more complex activations. Can the authors please clarify the architecture of the baseline? Regardless, as pointed out in the weaknesses above, it would really help the paper to have a more thorough empirical validation, for example, using deeper networks.

---

### Note · Authors · 2025-01-01

I have read and agree with the venue's withdrawal policy on behalf of myself and my co-authors.